# Genomic, Molecular, and Phenotypic Characterization of *Arthrobacter* sp. OVS8, an Endophytic Bacterium Isolated from and Contributing to the Bioactive Compound Content of the Essential Oil of the Medicinal Plant *Origanum vulgare* L.

**DOI:** 10.3390/ijms24054845

**Published:** 2023-03-02

**Authors:** Giulia Semenzato, Sara Del Duca, Alberto Vassallo, Angela Bechini, Carmela Calonico, Vania Delfino, Fabiola Berti, Francesco Vitali, Stefano Mocali, Angela Frascella, Giovanni Emiliani, Renato Fani

**Affiliations:** 1Department of Biology, University of Florence, Via Madonna del Piano 6, 50019 Sesto Fiorentino, Italy; 2Council for Agricultural Research and Economics, Research Centre for Agriculture and Environment, Via di Lanciola 12/A, 50125 Cascine del Riccio, Italy; 3School of Biosciences and Veterinary Medicine, University of Camerino, Via Gentile III Da Varano 1, 62032 Camerino, Italy; 4Department of Health Sciences, University of Florence, viale G.B. Morgagni, 48, 50134 Firenze, Italy; 5Institute for Sustainable Plant Protection (IPSP)—National Research Council (CNR), Via Madonna del Piano 10, 50019 Sesto Fiorentino, Italy

**Keywords:** plant microbiota, endophytes, volatile organic compounds, genome, essential oil

## Abstract

Medicinal plants play an important role in the discovery of new bioactive compounds with antimicrobial activity, thanks to their pharmacological properties. However, members of their microbiota can also synthesize bioactive molecules. Among these, strains belonging to the genera *Arthrobacter* are commonly found associated with the plant’s microenvironments, showing plant growth-promoting (PGP) activity and bioremediation properties. However, their role as antimicrobial secondary metabolite producers has not been fully explored. The aim of this work was to characterize the *Arthrobacter* sp. OVS8 endophytic strain, isolated from the medicinal plant *Origanum vulgare* L., from molecular and phenotypic viewpoints to evaluate its adaptation and influence on the plant internal microenvironments and its potential as a producer of antibacterial volatile molecules (VOCs). Results obtained from the phenotypic and genomic characterization highlight its ability to produce volatile antimicrobials effective against multidrug-resistant (MDR) human pathogens and its putative PGP role as a producer of siderophores and degrader of organic and inorganic pollutants. The outcomes presented in this work identify *Arthrobacter* sp. OVS8 as an excellent starting point toward the exploitation of bacterial endophytes as antibiotics sources.

## 1. Introduction

Antimicrobial resistance is a worldwide issue associated with high morbidity and mortality. Conventional antimicrobials are no longer able to overcome bacterial pathogens with a multidrug-resistant profile, resulting in difficult-to-treat or even untreatable infections [1]. The insufficiency of effective drugs and inefficient prevention measures have engaged prescribers and researchers in the development of novel treatment possibilities and alternative antimicrobial therapies [1].

In this context, medicinal plants can play an important role in the discovery of new natural bioactive compounds with antimicrobial activity. For thousands of years, plants have been the primary source of compounds with biological activities. Medicinal and aromatic plants have been widely used for the treatment and the prevention of various human diseases, such as cardiovascular, neurodegenerative, and hepatic diseases, skin disorders, and bacterial and viral infections [2,3,4,5,6]. In the last few decades, the research on natural products of plant origin has been almost fully replaced by drug design and combinatorial chemistry due to the time- and resource-consuming procedures that are required for the isolation, identification, and development of a product with pharmaceutical applications [7]. However, while chemical synthesis creates random molecules, plant secondary metabolites are produced as a result of an adaptation process and might be more efficient as therapeutic molecules in the battle against multidrug-resistant pathogens [7].

These bioactive compounds can also strongly affect the plant-associated microbiome and its physiological traits. It has been shown that the composition and structure of the microbial communities of the roots and the rhizosphere microenvironments change in response to the production of specific root exudates secreted by the plant as a result of microorganisms exposure [8,9,10]. Moreover, there is evidence of the different distribution of bacterial communities inside the internal tissues of the same medicinal plants, suggesting the existence of selective forces able to compartmentalize the microbes into specific microenvironments, which have been attributed to antagonistic interactions between microorganisms and their specific adaptation to the plant’s secondary metabolites [11,12,13,14,15]. Large-scale comparative genomic analyses also revealed that plant-associated bacteria have gained specific metabolic attributes which allow their adaptation to plant anatomical parts [16]. However, how secondary metabolites influence the relative abundance of a particular bacterial genus/species/strain and model the microbial community remains unclear.

A large number of medicinal plants have been characterized for their pharmacological properties and applications, but members of their microbiota are also able to synthesize bioactive molecules, both soluble and volatile ones [11,14,17,18,19]. Plants rely on their microbiome for specific traits, including nutrient acquisition, growth promotion, induced systemic resistance, and tolerance to abiotic stress factors [20]. Endophytes are considered an almost untapped source of natural compounds with potential therapeutical properties. Egamberdieva et al. [21] showed that plants whose extracts exhibited the highest antibacterial properties also hosted bacterial endophytes able to exert a comparable action toward bacterial pathogens. This confirms the evolutionary adaptation of endophytes to the plant microenvironments and identifies medicinal plants as an excellent starting point to isolate microorganisms able to produce biologically active molecules.

Among these bacteria, strains belonging to the genera *Arthrobacter*, Gram-positive bacteria of the family Micrococcaceae, are commonly found associated with the plant’s microenvironments: they are regular inhabitants of the rhizosphere [22,23], but they have also been isolated from the surface [24] and the inner tissues [25] of plants. Bacteria of the *Arthrobacter* genus often establish a beneficial relationship with their host plants through various mechanisms of action. Many *Arthrobacter* strains fulfill a plant growth-promoting activity, mostly due to the production of indole-3-acetic acid and siderophores, which is able to increase the solubility of various minerals and enhance nutrient acquisition [22,23,26]. In addition, many works focused on the interesting ability of *Arthrobacter* species to degrade organic and inorganic compounds which they utilize as substrates for their metabolism, thus making *Arthrobacter* genus a promising tool for plant-based bioremediation [24,27]. The *Arthrobacter* genus can be defined as a metabolically versatile group of bacteria, but their role as antimicrobial secondary metabolite producers has not been fully explored. Some plant-associated *Arthrobacter* strains, however, were able to antagonize both plant and human pathogens [28,29,30]. Taken together, these features highlight the importance of *Arthrobacter* strains from an ecological perspective and call attention to the possible biotechnological and therapeutical application of this bacterial genus.

In a recent work on *Origanum vulgare* L., the culturable endophytic microbiota isolated from different anatomical parts of the plant was characterized through molecular and phenotypic analysis, revealing a high degree of biodiversity at the strain level, different antibiotic resistance phenotypes, and the ability, for some strains, to antagonize the growth of bacteria isolated from the same or different *O. vulgare* microenvironments and some human pathogens [14]. Among others, *Arthrobacter* sp. OVS8, isolated from the stem of *O. vulgare* L., was able to inhibit the growth of various endophytes isolated from different compartments of the plant. Moreover, it efficiently inhibited the growth of ten members of the *Burkholderia cepacia* complex [14]. The antibacterial activity was attributed to the production of volatile organic compounds (VOCs): it has been demonstrated that some *O. vulgare* endophytes, including *Arthrobacter* sp. OVS8, synthesize a plethora of VOCs, some of which belong to the content of *O. vulgare* essential oil (EO), opening the possibility that (at least) some of the bioactive compounds of the plant EO might be synthesized by the microbiota itself [19].

A better understanding of the pathways and genes involved in the biosynthesis of endophytic secondary metabolites and of the role of such compounds in the interplay between the plant and its associated microbial communities should be strongly encouraged to unravel the mechanisms implied in the plant–microbiota interactions and so enable the prediction of endophytes’ capacity to synthesize novel bioactive secondary metabolites and potential antibacterial drug candidates [31]. Thus, this work aims to delineate molecular and phenotypic features of *Arthrobacter* sp. OVS8 from different viewpoints in order to evaluate its adaptation and influence on the plant internal microenvironments and its biotechnological potential as a producer of antibacterial volatile molecules.

## 2. Results

### 2.1. Diesel Fuel- and β-Caryophyllene-Degrading Potential

To examine its adaptation to the plant essential oil, *Arthrobacter* sp. OVS8 was tested for its ability to grow in the presence of diesel fuel and β-caryophyllene as the only carbon and energy sources.

The endophyte growth in the presence of diesel fuel was evaluated as the variation of the density of the bacterial streak, measured from 4 (complete growth, as the positive control) to 0 (absence of growth). Its growth was maximal (4) on the Tryptic Soy Agar (TSA) control plate and on Minimum Davis medium (MMD) supplemented with glucose (4), and it was strong (3) on MMD supplemented with diesel fuel.

We then assessed the ability of *Arthrobacter* sp. OVS8 to resist in the presence of β-caryophyllene and/or use it as a carbon and energy source. This sesquiterpene hydrocarbon is one of the main components of *O. vulgare* essential oil [14]. The strain was plated in the presence of a 6 mm paper disk soaked with different concentrations of the volatile compound, as described in the Materials and Methods section. If the strain can degrade/utilize β-caryophyllene, its growth on minimum medium would only be possible in the proximity of the paper disc soaked with the compound. The experiment was repeated twice, and results are reported in Table 1. *Arthrobacter* sp. OVS8 was sensible to β-caryophyllene, as its growth was inhibited on Mueller Hinton (MH) agar and on MMD supplemented with glucose, while it did not grow in the presence of β-caryophyllene as the only source of carbon and energy.

### 2.2. Antibacterial Properties of Arthrobacter sp. OVS8 VOCs against MDR Human Pathogens

Since we previously demonstrated that *Arthrobacter* sp. OVS8 synthesize VOCs able to completely inhibit the growth of bacteria belonging to the *Burkholderia cepacia* complex (Bcc) [19], it was tested for its ability to produce VOCs able to antagonize the growth of 36 multidrug-resistant human pathogenic strains belonging to the *Staphylococcus aureus*, *Pseudomonas aeruginosa*, *Klebsiella pneumoniae* and Coagulase-Negative Staphylococci (CoNS) groups. The analysis was carried out through the cross-streaking method, using Petri dishes with a septum physically separating the culture plate into two compartments, as described in Materials and Methods. The antibacterial activity of *Arthrobacter* sp. OVS8 was evaluated qualitatively as the reduction in the density of the target strain streak, measured from 0 (where the strain grew as the positive control) to 4 (absence of growth). The obtained data are shown in Table 2.

Of the 36 target strains, *Arthrobacter* sp. OVS8 VOCs were able to slightly reduce the growth of 7 strains, belonging to the *P. aeruginosa* and *S. aureus* groups, while it exhibited a moderate antagonistic effect toward 4 *K. pneumoniae* strains and 1 of the CoNS strains, namely, *S. epidermidis* 5321.

### 2.3. Quantification of VOCs Activity against Klebsiella pneumoniae Strains

Given the results obtained through the qualitative cross-streaking, we decided to focus on the *K. pneumoniae* group. The inhibitory activity of the VOCs emitted by *Arthrobacter* sp. OVS8 was quantified through a quantitative cross-streaking method, which allows the determination of the number of viable target strain cells at the beginning (t_0_, soon after their streaking onto plates) and at the end of the experiment (t_1_, after 48 h incubation in the presence or absence of the tester strain) [19]. The experiment was performed in triplicate according to the protocol detailed in the Materials and Methods section.

The obtained results are schematically represented in Figure 1 and Figure 2. *Arthrobacter* sp. OVS8 was able to significantly reduce the growth of all the *K. pneumoniae* strains (Figure 1A and Figure 2); indeed, the *t*-test between the average CFU obtained at t_1_ (normalized for the initial titer) for treated and non-treated *K. pneumoniae* strains highlighted a significant difference between the two conditions (*p*-value < 0.005). Among all the strains, *K. pneumoniae* ATCC 700603 resulted to be the least sensitive target for the VOCs emitted by the endophyte. The strongest antibacterial activity was reported against *K. pneumoniae* 4409 and 4420, with a CFU reduction higher than 50% compared with the growth controls (Figure 1B).

### 2.4. Genome Analysis of Arthrobacter sp. OVS8

Data regarding the genome assembly of the sequencing reads performed through the Canu assembler plugin is shown in Table 3. *Arthrobacter* sp. OVS8 embeds a single contig with an overall length of 4,175,013 bp and a GC content of 67.14%. The genome was annotated using the NCBI GenBank annotation pipeline, revealing the presence of 4273 genes, of which 2598 are coding genes, 15 are rRNAs, 50 are tRNAs, and 3 are ncRNAs. The complete genome sequence is available in GenBank under the accession number CP116670.

We also performed an Average Nucleotide Identity (ANI) and a digital DNA–DNA hybridization (dDDH) analyses, as described in Materials and Methods. Data obtained are reported in Appendix A and revealed that this strain might potentially be affiliated to a new species.

### 2.5. Secondary Metabolite Biosynthetic Gene Clusters

Secondary metabolite biosynthetic gene clusters (BGCs) analysis predicted five types of BGCs (Table 4); however, only one BGC had a percentage of similarity higher than 50% with clusters available in the database. This BGC accounts for the production of desferrioxamine, a non-ribosomal peptide-synthetase (NRPS)-independent siderophore; a 66% of similarity was found with the desferrioxamine BGC from *Streptomyces argillaceus*.

### 2.6. Metabolic Pathways Involved in the Synthesis of Arthrobacter sp. OVS8’s Emitted VOCs

The DuctApe software suite is able to handle annotated genomic data, enabling metabolism reconstruction according to the KEGG database: each protein that has a KEGG annotation in the genome is mapped to its KEGG reaction ID and the corresponding pathway [32].

Given the previous data showing that *Arthrobacter* OVS8-emitted VOCs are able to induce a bactericidal effect against some human opportunistic pathogens belonging to the Bcc, we focused on the metabolic pathways involved in the production of such VOCs; in particular, to those which are also part of *O. vulgare* essential oil composition, i.e., α-pinene, p-cymene, and γ-terpinene [19]. The biosynthesis of these volatile compounds, all belonging to the monoterpenes family, contemplates the catabolic activity of various terpene synthases and begins with the ionization of the geranyl diphosphate [33]. As depicted in Figure 3, *Arthrobacter* sp. OVS8 shows the metabolic potential to produce geranyl diphosphate. However, no enzyme belonging to the subsequent monoterpenoid biosynthetic pathway was identified in the annotated genome of *Arthrobacter* sp. OVS8 (Figure 4).

### 2.7. Phenotype Microarray Results

The phenome of *Arthrobacter* sp. OVS8 was investigated using the GEN III aerobe microbial identification plates. Data were analyzed with the DuctApe software suite [32] to obtain activity values (AVs) for each well. The AV is a synthetic parameter able to recapture the whole shape of the respiration curve, accounting for five parameters of the curve, namely, the length of the lag phase, the slope, the average height, the maximum cell respiration, and the area under the curve.

#### 2.7.1. Growth of *Arthrobacter* sp. OVS8 under Different Stresses

The GEN III microplate contains 23 chemical sensitivity assays. Based on the AV, we may classify *Arthrobacter* sp. OVS8 as being able or not to grow under different stresses. For exploration, samples with an AV lower than seven were grouped as having low or absence of growth (i.e., no resistance to the harmful molecule in the sensitivity assay), while those with AVs equal to or higher than seven were considered high growth (i.e., showing resistance to the harmful molecule in the sensitivity assay). Regarding chemo-physical stress assays, *Arthrobacter* sp. OVS8 was able to grow in the presence of NaCl concentrations up to 4% but showed no growth at 8% NaCl, and it was able to grow at a pH of 6 but showed no growth at pH 5 (Figure 5).

Antibiotic resistance of *Arthrobacter* sp. OVS8, based on the antibiotics included in the GEN III plate, was fairly low. It showed resistance to nalidixic acid and aztreonam, but sensitivity to troleandomycin, rifamycin SV, minocycline, lincomycin, and vancomycin. For other classes of chemical compounds, *Arthrobacter* sp. OVS8 displayed resistance to 1% sodium lactate, d-serine, lithium chloride, potassium tellurite, and sodium bromate, while it showed sensitivity to fusidic acid, guanidine hydrochloride, Niaproof 4, tetrazolium violet, and tetrazolium blue (Figure 5 and Table 5).

#### 2.7.2. Carbon and Nitrogen Source Utilization of *Arthrobacter* sp. OVS8

The same approach used for the evaluation of results on the sensitivity wells was also used for the C- and N-utilization wells. Samples with AV lower or equal to three were considered as showing absence of growth (i.e., inefficient utilization of the C or N source), while those with AV equal to or higher than seven were considered as high growth (i.e., efficient utilization of the C or N source). Overall, *Arthrobacter* sp. OVS8 was able to efficiently utilize 23 out of 68 C sources included in the GEN III microplate, while showing low or no growth in the presence of 27 out of 68 C sources included in the GEN III microplate (Figure 5 and Table 6). In addition, *Arthrobacter* sp. OVS8 showed modest utilization of 18 C sources included in the GEN III microplate, showing AVs between 3 and 7.

As far as nitrogen utilization is concerned, the GEN III microplates include a total of four wells with different N sources. *Arthrobacter* sp. OVS8 efficiently utilized the g-amino-N-butyric acid, but inefficiently utilized D-aspartic acid, glycil-L-proline, and L-aspartic acid.

## 3. Discussion

The lack of new compounds able to combat the current issues of antibiotic resistance, along with the emergence of multidrug-resistant pathogens and the continued presence of untreatable diseases, currently represent a challenge for the scientific community in the study of new and effective molecules with antibacterial activity. In fact, only a small fraction of the antibiotics approved over the past 40 years represent new compound classes, while the majority are derived from already known chemical structures, with the most recent new class of antibiotics being discovered during the 1980s [34]. Over the last few decades, and more recently over the SARS-COVID19 pandemic, there has been a remarkable resurgence of interest in natural product research to both prevent the toxicity effects of some medical products and to discover new antibiotic-producing strains [35]. When speaking about natural products, traditional medicine, also known as alternative or complementary medicine, cannot be ignored. The herbs, plants or formulas used in traditional medicine contain a plethora of phytochemicals that function alone or in combination with one another to produce a pharmacological effect. Indeed, many plant-originated drugs were discovered from traditional medicine knowledge, and it has been demonstrated that many of them were discovered thanks to their application in such practices [36].

The renewed interest in medicinal plants and their bioactive secondary metabolites has also put the spotlight on the complex and multifaced world of bacterial endophytes and their intimate and intricate relationship with their hosts. There is evidence of endophytes adapting to the plant microenvironments and their direct or indirect contribution to plant secondary metabolites [37,38,39], so one may ask (i) if the phytochemicals obtained from medicinal plants could be a result of the bacterial metabolisms, and (ii) if such bioactive compounds could be directly obtained from endophytes.

In this work, we performed a genomic, molecular, and phenotypic characterization of *Arthrobacter* sp. OVS8, a promising bacterial endophyte isolated from the medicinal plant *O. vulgare* L. [14]. In particular, we focused on specific features of the strain: its adaptation to the plant microenvironment and essential oil and its ability to inhibit the growth of some multidrug-resistant bacterial pathogens through the emission of VOCs.

*O. vulgare* essential oil is widely known for its antimicrobial potential [40]; thus, it can be imagined that bacterial endophytes inhabiting the inner microenvironment of such plants might be adapted to the bioactive volatile molecules of which the plant tissues are rich in [12], becoming resistant to them or using them as a carbon and energy source. Shimasaki et al. revealed that tobacco roots, a microenvironment abounding in nicotine and santhopine, were enriched in *Arthrobacter* strains with the catabolic capacity to detoxify or utilize them as nutrients, further supporting the relevance of plant secondary metabolites in the shaping of the plant’s microbial community with specific metabolic competences [41]. The diesel fuel-degrading potential test revealed the ability of *Arthrobacter* sp. OVS8 to grow in the presence of diesel fuel as the only carbon and energy source. The GC/MS analysis of the essential oil hydro-distilled from the same *O. vulgare* plant from which the endophyte was isolated revealed that the major chemical groups consisted of sesquiterpene hydrocarbons (73.5%), monoterpene hydrocarbons (17.6%), oxygenated sesquiterpenes (4.8%), and 3.7% oxygenated monoterpenes (4.8%) [14]. The ability of the endophyte to grow in the presence of diesel fuel could suggest its adaptation to the hydrocarbon components of *O. vulgare* essential oil, leading to the hypothesis that its composition might represent one of the factors involved in the plant–endophytes symbiosis [13]. For this reason, we set up another experiment using β-caryophyllene, a bicyclic sesquiterpene representing 19.2% of *O. vulgare* essential oil, as the only carbon and energy source. Sesquiterpenes are a class of terpenes that consist of three isoprene units. *Arthrobacter* strains isolated from the under-canopy soil of the isoprene-emitting *Tectona grandis* and *Madhuca latifolia* trees had a high isoprene tolerance and a great degrading potential toward the compound [42]. Our results suggest that β-caryophyllene does not represent a carbon or energy source for *Arthrobacter* sp. OVS8. Moreover, according to Ponce et al., the strain can be classified as extremely sensitive (inhibition zone diameter ≥ 20 mm) to β-caryophyllene (100%) [43]. As the strain was isolated from the stem of *O. vulgare*, it can be hypothesized that the content and composition of essential oil from flowers, leaves and stems differ among anatomical parts [44] or that the endophyte is not intimately associated with plant organs and cells responsible for the production of the essential oil [45]. The hydrocarbon-degrading potential of the endophytes could be then associated with an adaptive mechanism through which the endophyte takes advantage of the abundant long chain aliphatic compounds that constitute the plant tissues, as hypothesized for epiphytic bacteria [46].

*Arthrobacter* sp. OVS8 was selected amongst the collection of endophytes isolated from *O. vulgare* because of its antibacterial activity against 10 members of the *Burkholderia cepacia* complex (Bcc), opportunistic pathogens able to cause severe infections in immunocompromised patients, such as cystic fibrosis patients. This activity was attributed also to the production of VOCs, which induced a bactericidal effect toward 7 out of 10 Bcc target strains, most of which were of clinical origin [19]. The endophyte’s emitted VOCs were tested against a panel of 36 multidrug-resistant human pathogens through the cross-streaking method. Data obtained revealed that target strains used in this work were much more resistant than Bcc strains, and only *K. pneumoniae* strains’ growth seemed to be affected by *Arthrobacter* sp. OVS8 VOCs. There is still little evidence on the antibacterial metabolites produced by *Arthrobacter* species, and most of them refer to strains isolated from the Antarctic or marine environments [47,48]. Further strategies for the identification of new antibiotics produced by unexplored targets (such as bacterial endophytes associated with medicinal plants) are required to ensure the availability of effective drugs and overcome the issue of antimicrobial resistance.

The genome of *Arthrobacter* sp. OVS8 has a total length of 4,175,013 bp and a GC content of 67.14%, which reflects the characteristic high GC content of the genus. Secondary metabolite biosynthetic gene clusters (BGCs) analysis predicted a desferrioxamine BGC, a NRPS-independent siderophore, with a 66% similarity to the desferrioxamine BGC from *Streptomyces argillaceus* [49]. The production of desferrioxamine is quite common for species belonging to the *Streptomyces* genus, and it benefits the plant–microorganism interaction by being able to promote plant growth, alleviate oxidative stress, and promote the solubilization of iron and other metals [50,51,52]. Siderophore-producing bacteria play a crucial role in plants’ survival and growth, enhancing metal bioavailability in the rhizosphere [53]. Siderophore production has also been reported for the genus *Arthrobacter*. A high siderophore-yielding *Arthrobacter* strain isolated from the wild grass *Dichanthium annulatum* colonizing an abandoned mine efficiently solubilized iron and increased iron-stress resilience in iron-deficiency-sensitive maize [26]. Moreover, the naturally occurring rhizosphere bacteria *Arthrobacter oxydans* releases a variety of desferrioxamine-like compounds that induce direct plant growth-promoting effects and increase the mobility and solubility of a great variety of metals and minerals through the formation of soluble element-organic complexes that can move toward the plant roots [23]. Hence, the presence of the desferrioxamine BGC in the *Arthrobacter* sp. OVS8 genome could suggest a putative plant-growth promoting role of the endophyte in facilitating *O. vulgare* nutrient acquisition.

Given the previous data regarding the ability of *Arthrobacter* sp. OVS8-emitted VOCs to induce a bactericidal or bacteriostatic effect against some clinical and environmental Bcc isolates, we focused on the metabolic pathways involved in the production of such VOCs; in particular, to three monoterpenes also found in *O. vulgare* essential oil composition, i.e., α-pinene, p-cymene, and γ-terpinene [19]. In nature, we can find different terpene carbon skeletons. Such abundance is attributed to the large number of the enzyme class of terpene synthases and to their ability to convert the same substrate to multiple products [33]. Even though there is a solid degree of amino acid sequence similarity among plant and fungi monoterpene synthases, this similarity is based more on taxonomic affinities of the plant species rather than the type of compound formed. The best recognized structural motif of the terpene synthase family is an aspartate-rich region, [D/N)DXX(D/E) or DDXXXE], located within 80–120 (bacteria and fungi) or 230–270 aa (plants) of the N terminus [33,54]. However, the discovery and biochemical characterization of bacterial terpene synthases represent a great challenge because, unlike plant and fungal enzymes, bacterial terpene synthases do not exhibit an overall amino acid sequence similarity to those from plants and fungi and display a relatively low degree of sequence similarity to other known bacterial terpene synthases [54]. Through the use of hidden Markov models and protein family database searching methods, various structurally diverse bacterial terpene synthases were identified; most of these were sesquiterpene synthases widely distributed amongst Gram-negative bacteria and the *Actinomycetales* order, as the geosmin producer *Streptomyces* genus [54,55]. The genome analysis of *Arthrobacter* sp. OVS8 revealed the presence of the metabolic pathway leading to the synthesis of geranyl diphosphate, the common precursor of most of the cyclic monoterpenes; however, no monoterpene biosynthetic pathways were highlighted. Given the fact that the endophyte actually produces monoterpenes [19], the absence of the identification of the latter pathway could be attributed to the high diversity existing between the amino acid sequence of plant and bacteria terpene synthases and the lack of bacterial terpene synthase sequences from the *Arthrobacter* genus.

GEN III microplates dissect and analyze the ability of strains to metabolize all major classes of compounds and determine other important physiological properties such as pH, salt, and lactic acid tolerance, reducing power, and chemical sensitivity. It could be assumed that strains metabolizing a wide range of substrates and showing tolerance to high salinity concentrations and acidic pH could be more adapted to changing environmental conditions. *Arthrobacter* sp. OVS8 was able to grow in a medium containing up to 4% NaCl and a pH of 6, suggesting its ability to tolerate mild changes in abiotic stress factors. The strain showed a high resistance toward 10 out of 23 available chemical sensitivity assay compounds such as D-serine, which exerts its antimicrobial activity by replacing D-alanine residues of the peptidoglycan rigid envelope surrounding the cytoplasmic membrane of bacterial species [56]. Resistance to sodium lactate, a salt of lactic acid widely used as a food preservative able to lower water activity [57], and to lithium chloride, reported to induce hyperosmotic stress [58], could indicate the endophyte’s potential to adapt to changes in water availability and high saline concentration. Potassium tellurite toxicity is due to its ability to act as a strong oxidizing agent over a variety of cell components [59]. *Arthrobacter* sp. OVS8 growth in the presence of such compounds could hint at the endophyte’s ability to remove toxic tellurite from polluted environments [60]. Lastly, *Arthrobacter* sp. OVS8 resistance to sodium bromate, which mainly originates when ozonation is adopted to treat bromide (Br−)-containing water, could suggest its role as a potential candidate for bioremediation processes [61]. On the contrary, the strain did not grow in the presence of the disinfectant polyhexamethylene guanidine hydrochloride, which disrupts the cellular envelope causing leakage of the cytoplasmic content [62], and the surfactant Niaproof 4. *Arthrobacter* sp. OVS8 was sensitive to most of the antibiotics present in GEN III plates: the protein synthesis inhibitors fusidic acid, troleandomycin, rifamycin, minocycline, and lincomicyn, as well as vancomycin, which hampers proper cell wall synthesis in Gram-positive bacteria, did not permit the growth of the endophytic strain. As a Gram-positive bacterium, *Arthrobacter* sp. OVS8 showed resistance in the presence of nalidixic acid and aztreonam, antimicrobial molecules more effective toward Gram-negative strains [63,64].

## 4. Materials and Methods

### 4.1. Bacterial Strains and Growth Conditions

*Arthrobacter* sp. OVS8 was isolated from the stem of the medicinal plant *O. vulgare* L. as described in Castronovo et al. [14]. This strain belongs to a collection of isolates obtained from a pool of *O. vulgare* plants cultivated in a common garden at the “Giardino delle Erbe”, Casola Valsenio (Italy). *Arthrobacter* sp. OVS8 was maintained on Tryptic Soy Agar (TSA, Biolife, Milan, Italy) plates for 48 h at 30 °C.

*Arthrobacter* sp. OVS8 was tested against 36 pathogenic strains: 10 CoNS, 10 *P. aeruginosa*, 10 *S. aureus*, and 6 *K. pneumoniae* strains, all characterized by their resistance to multiple antibiotics (as reported in Table 7) [65]. The strains were grown on TSA plates at 37 °C for 24 h.

### 4.2. Growth in the Presence of Diesel Fuel and O. vulgare Essential Oil Main Compounds

The endophytic strain was tested for its ability to grow in the presence of diesel fuel as the sole carbon and energy source. A single colony of *Arthrobacter* sp. OVS8 was suspended in 100 μL of 0.9% *w/v* NaCl solution and then streaked on minimal medium Davis [66] (MMD, 1 g (NH_4_)_2_SO_4_, 7 g K_2_HPO_4_, 2 g KH_2_PO_4_, 0.5 g Na3-citrate·2H_2_O, 0.1 g MgSO_4_·7H_2_O, pH 7.2, per liter of deionized water) containing 0.4% *v/v* diesel fuel or 1% *w/v* glucose as the sole carbon and energy source. Diesel fuel (Esso Italiana, Roma, Italy) was previously filtered through a 0.2 μm-pore-size filter (Sartorius) for sterilization and particle removal. TSA plates were used as growth control. Once streaked, the endophytes were incubated at 30 °C for 3 days. Positivity to this assay was assessed as the presence or the absence of visible growth, expressed as a range from 4 (complete growth) to 0 (absence of growth).

The strain was also tested for its ability to grow in the presence of β-caryophyllene as the sole carbon and energy source. A few colonies of the strain were inoculated in 10 mL of Tryptic Soy Broth (TSB, Biolife) and incubated at 30 °C overnight. A dilution of the inoculum was prepared (OD600 = 1) and 100 μL of such suspensions was spread onto different types of agar media: Muller–Hinton Agar (MH, 17.0 g/L agar, 2 g/L beef heart infusion, 17.5 g/L casein acid hydrolysate, 1.5 g/L starch), MMD supplemented with 1% of glucose, and MMD. Immediately after the spreading, sterile filter paper disks (Oxoid S.p.A. Milan, Italy) of 6 mm diameter were placed on the surface of the dishes and soaked with 10 µL of β-caryophyllene and with either a 1:10 or 1:100 dilution of the compound in dimethyl sulfoxide (DMSO) 0.5% *v/v*. In addition, positive and negative controls were prepared using the antibiotic chloramphenicol (1 mg/mL) (Oxoid S.p.A.) and a solution of DMSO 0.5% in sterile deionized water, respectively.

The plates were checked after a 48 h incubation at 30 °C and the diameter of the inhibition zones, including the paper disc diameter, was measured in mm.

### 4.3. Evaluation of the Antibacterial Activity of Bacterial Volatile Organic Compounds against Human Pathogens

*Arthrobacter* sp. OVS8 antibacterial activity was evaluated through cross-streaking, using Petri dishes with a septum separating two compartments to allow the growth of the tester and the target strains without any physical contact. A few colonies of *Arthrobacter* sp. OVS8 were suspended in 2 mL of a 0.9% NaCl *w/v* solution, reaching a McFarland turbidity standard of 0.5, corresponding to 1 × 10^7^ CFU/mL. Using a cotton swab, the obtained suspension was streaked on one of the two halves of the Petri dishes. Plates were then incubated at 30 °C for 48 h to allow growth of the strain and the production of volatile organic compounds. A single colony of each target strain was then streaked perpendicularly to the septum using an inoculation needle, and plates were incubated at 37 °C for a further 48 h. Additionally, target strains were streaked on half of a Petri plate in the absence of the tester as a growth control. The antagonistic effect was evaluated at 24 and 48 h and measured as the reduction of the target strains’ growth in the presence of the tester compared to control plates. The different inhibition levels were indicated as follows: complete (4), strong (3), moderate (2), weak (1), and absent (0) inhibition.

### 4.4. Quantification of VOCs Activity against Klebsiella Pneumoniae Strains

The antibacterial activity of VOCs synthesized by *Arthrobacter* sp. OVS8 was quantified through the quantitative cross-streaking method described in Polito et al., 2022 [19], with some modifications. Briefly, a few *Arthrobacter* sp. OVS8 colonies were suspended in 2 mL of a 0.9% NaCl *w/v* solution, reaching a McFarland turbidity standard of 0.5. Using a cotton swab, the obtained suspension was streaked on one of the two halves of the Petri dishes. Plates were incubated at 30 °C for 48 h. After two days, a few colonies of each target strain (previously grown at 37 °C for 24 h on TSA plates) were suspended in 2 mL of a 0.9% NaCl *w/v* solution in order to obtain turbidity corresponding to a 0.5 McFarland standard; then, a 1:100 dilution was prepared. With the aid of a micropipette, 30 μL of the obtained suspensions were applied on the second half of the Petri plate, near the septum. The six drops were then streaked perpendicularly to the septum with an inoculation needle. For the growth control, the same procedure was repeated on Petri plates in the absence of the tester strain. The target suspensions were spread on a TSA plate, using appropriate dilutions to obtain the number of viable cells streaked at the beginning of the experiment (t_0_). All these plates were incubated at 37 °C for 48 h days.

To determine the number of target strains’ viable cells grown in the presence and in the absence of *Arthrobacter* sp. OVS8 (t_1_), target cells were recovered in 2 mL of saline solution with a spatula. The suspensions obtained were appropriately diluted, spread onto TSA plates, and incubated at 37 °C for 24 h to determine the viable titer. The number of CFUs obtained at t_1_ in the presence of *Arthrobacter* sp. OVS8 was then compared with those obtained for the growth control in the absence of the tester strain and with the number of viable cells streaked at the beginning of the experiment (t_0_).

### 4.5. Phenotype Microarray

Phenotype microarray analysis was performed using the Biolog GEN III MicroPlate^TM^ (Catalog No.1030, Biolog). The test provides a phenotypic fingerprint of the microorganism as it contains 71 carbon sources and 23 chemical sensitivity assays. All the reagents used were provided by Biolog, Inc. (Hayward, CA, USA). *Arthrobacter* sp. OVS8 was grown on TSA at 30 °C for 48 h. The bacterial suspension was prepared by picking bacterial colonies with a sterile inoculation loop and resuspending them in IF-B (Catalog No.72402, Biolog), until 98% transmittance was reached, as assessed with the Biolog turbidimeter (Catalog No.3587). Then, 100 µL of the obtained suspension was dispensed into each well of a Biolog GEN III MicroPlate^TM^. The plate was incubated at 30 °C in an Omnilog reader (Biolog) for 4 days. The data obtained was visualized using the Biolog Data Analysis software v.1.7. Phenomic data were analyzed using the DuctApe software suite (version 0.18.2) [32] to obtain the activity values (AVs) of each well, and plotted using the ggplate R package (https://github.com/jpquast/ggplate, https://jpquast.github.io/ggplate/, accessed on 30 January 2023).

### 4.6. DNA Extraction and Genome Sequencing

A single colony of *Arthrobacter* sp. OVS8 was inoculated in 10 mL of fresh Tryptic Soy Broth (Biolife) in a 50 mL tube and incubated at 30 °C overnight under shaking (130 rpm). The following day, bacterial cells were collected by centrifugation (15,500× *g* for 4 min) and the PowerLyzer PowerSoil DNA Isolation Kit (MO BIO Laboratories, Inc., Carlsbad, CA, USA) was used to extract genomic DNA, following the protocol provided by the manufacturer, with some modifications [31]. Specifically, the cell pellet was resuspended in the PowerSoil Bead Solution in the presence of 1 mg/mL of lysozyme and incubated for 1 h at 37 °C. PowerSoil Solution C1 and 0.5 mg/mL proteinase K were added to the sample, which was then incubated at 55 °C for 2 h before proceeding with the next DNA purification steps.

Nanopore sequencing was performed with a PCR-free approach following the native barcoding genomic DNA protocol provided by Oxford Nanopore Technologies (ONT) (version NBE_9065_v109_revY_14Aug2019), as described in Semenzato et al. (2022). The gDNA of *Arthrobacter* sp. OVS8 was sequenced as follows. Briefly, 1 µg of each input gDNA was repaired and end-prepped using the NEBNext Companion Module for Oxford Nanopore Technologies Ligation Sequencing (E7180S, New England Biolabs, MA, USA). Purification with Agencourt AMPure XP beads (Beckman Coulter, CA, USA) on a magnetic separator followed, and the concentrations of DNA samples were determined using a Qubit dsDNA HS Assay Kit and a Qubit 4 Fluorometer (ThermoFisher Scientific, MA, USA). Then, 500 ng of each end-prepped DNA sample was barcoded using NEB Blunt/TA Ligase Master Mix (M0367, New England Biolabs) and the Native Barcoding Expansion 13–24 (EXP-NBD114, ONT). After purification, equimolar amounts of barcoded DNA samples were pooled to obtain a total of 700 ng and subjected to the adapter ligation. During the subsequent clean-up step, the Long Fragment Buffer included in the Ligation Sequencing Kit (SQK-LSK109, ONT) DNA library was used to enrich the DNA library with >3 kb-long fragments, and it was immediately sequenced. A R9.4.1 Flow Cell (FLO-MIN106D, ONT) was primed with a Flow Cell Priming Kit (EXP-FLP002, ONT). The library was loaded following the instructions provided and sequencing was performed with a MinION MK1B (ONT) and the MinKNOW software v.21.10.4 for 72 h. Base calling and demultiplexing were performed using Guppy v.4.3.4.

### 4.7. Genome Assembly, Annotation and Bioinformatic Analysis

De novo assembly was achieved using Canu assembler software v.2.1.1 [67] and the quality of contigs was assessed by QUAST v.5.0.2 [68]. These functions were performed in a Galaxy environment (https://usegalaxy.eu, accessed on 3 September 2021). The assembled genome sequence was annotated using the NCBI Prokaryotic Genome Annotation Pipeline (PGAP) v.6.4 (https://www.ncbi.nlm.nih.gov/genome/annotation_prok/, accessed on 20 January 2023). The Average Nucleotide Identity (ANI) analysis was performed using FastANI v.1.3, with default options [69]. The genomic sequences of the genera closely related to the bacterial strain under investigation (*Arthrobacter* and *Pseudarthrobacter*) were downloaded from the NCBI “assembly” database and used as reference input for the ANI analysis.

The genome sequence of *Arthrobacter* sp. OVS8 was then uploaded to the Type (Strain) Genome Server (TYGS), a free bioinformatics platform available under https://tygs.dsmz.de, accessed on 30 January 2023, for a whole genome-based taxonomic analysis [70,71]. The results were provided by the TYGS on 22 February 2023.

### 4.8. Secondary Metabolite Biosynthetic Gene Clusters

The antiSMASH v.6.0.1 webserver was used for the identification of gene clusters involved in the biosynthesis of secondary metabolites [72]. The query genome was uploaded in a FASTA format; to identify only well-defined clusters containing genes with a significant alignment, the analysis was performed using the strict method of detection.

## 5. Conclusions

In conclusion, this work aimed at characterizing the newly isolated endophytic strain *Arthrobacter* sp. OVS8, isolated from the stem of the medicinal plant *O. vulgare*, to test its ability to synthesize antimicrobial compounds effective against bacterial human pathogens and its adaptation to plant microenvironments. For this purpose, a set of phenotypic and genetic parameters was tested. Cross-streaking experiments revealed that *Arthrobacter* sp. OVS8-emitted VOCs are able to antagonize the growth of different bacterial pathogens, especially *K. pneumoniae* strains, which exhibit a multi-drug-resistance phenotype. This is particularly intriguing, considering the worldwide spreading of pathogenic bacteria with multidrug-resistance profiles. *Arthrobacter* sp. OVS8 association with the plant internal tissues does not seem related to its ability to degrade and/or utilize the essential oil compound β-caryophyllene, but its hydrocarbon degrading potential might suggest an intimate relationship with the plant internal tissues. The phenotype microarray analysis revealed the capacity of *Arthrobacter* sp. OVS8 to grow in the presence of various chemicals, confirming its ecological role in the degradation of organic and inorganic compounds, thus representing a tool for bioremediation. Finally, genome analysis pointed out the production of siderophores, which might suggest its role as a plant-growth-promoter bacterium, while its genetic features regarding VOCs biosynthesis could not be fully elucidated. *Arthrobacter* sp. OVS8 represents an excellent starting point toward the exploitation of bacterial endophytes as antibiotics sources.

## Figures and Tables

**Figure 1 ijms-24-04845-f001:**
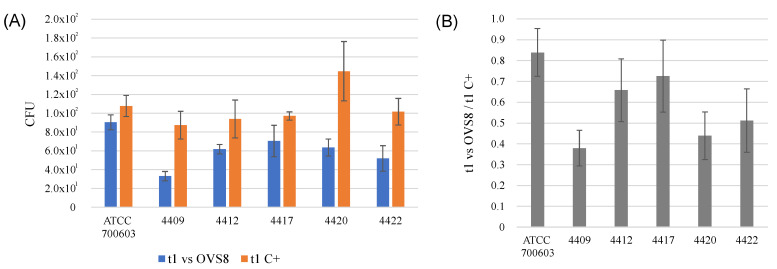
(**A**) Schematic representation of the average number of colony-forming units (CFU) of each target strain, normalized on the initial titer, in the absence (orange bar) or in the presence (blue bar) of the endophytic tester strain. (**B**) Ratio between the average CFU in the presence and absence of the endophytic tester strain. Black bars represent standard errors.

**Figure 2 ijms-24-04845-f002:**
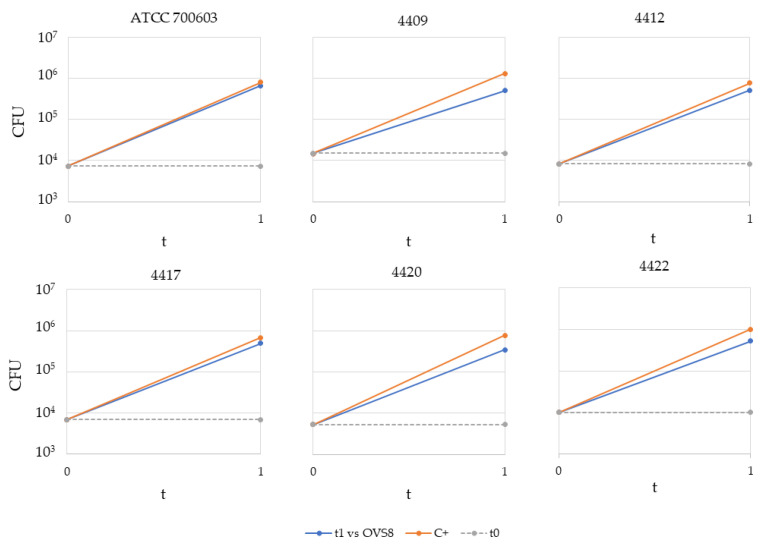
Effect of *Arthrobacter* sp. OVS8’s VOCs on growth of target strains.

**Figure 3 ijms-24-04845-f003:**
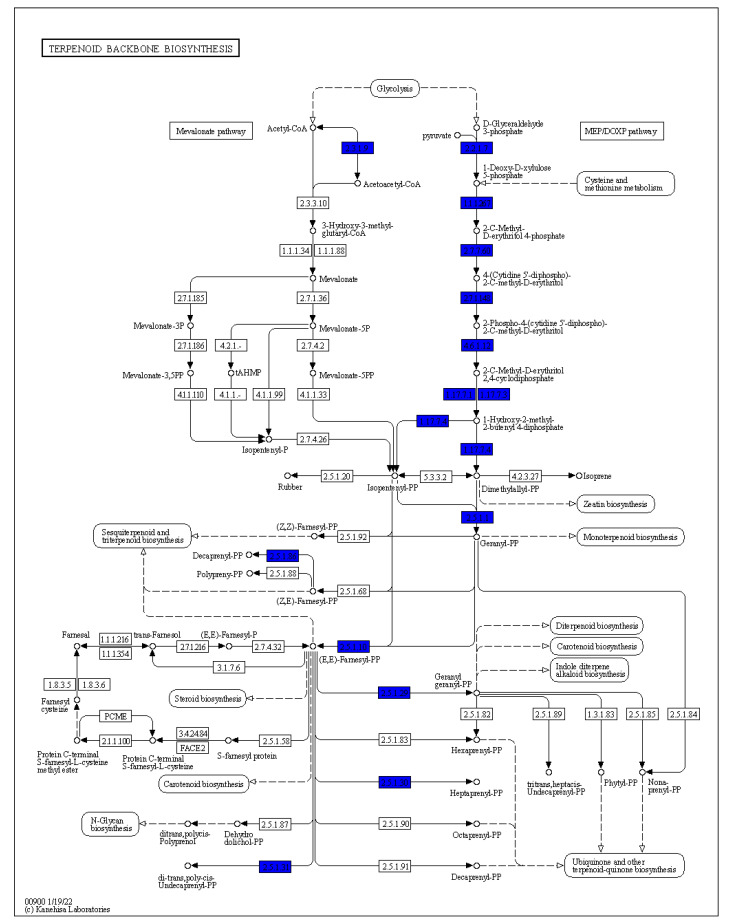
Terpenoid backbone biosynthesis KEGG map. Enzymes retrieved by the DuctApe software suite are reported in blue.

**Figure 4 ijms-24-04845-f004:**
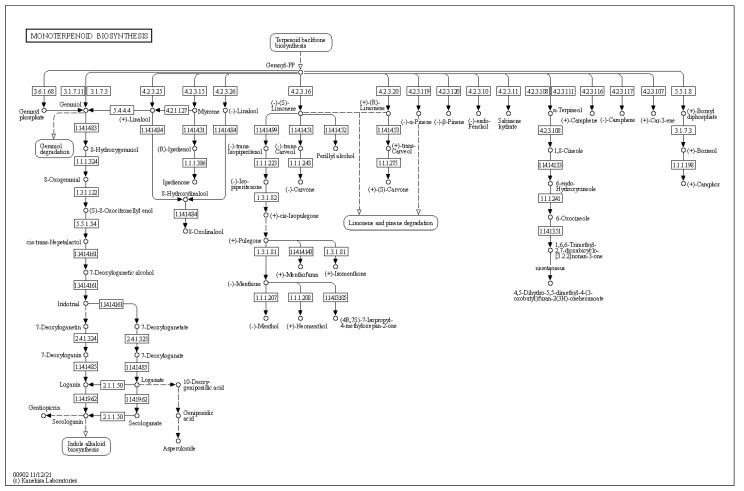
Monoterpenoid biosynthesis KEGG map. Enzymes retrieved by the DuctApe software suite are reported in blue.

**Figure 5 ijms-24-04845-f005:**
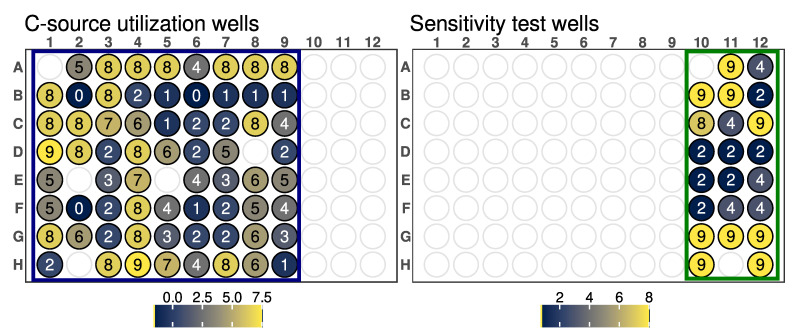
Schematic representation of the GEN III microplate of the Biolog system, divided into the C-source utilization (**left**) and sensitivity test (**right**) well subsets. Each well is colored based on the calculated AV, which is indicated by the reported number.

**Table 1 ijms-24-04845-t001:** Effect of β-caryophyllene on the growth of *Arthrobacter* sp. OVS8. Inhibition zone diameters are reported as the mean value of the two replicates.

Medium	Inhibition Zone (mm)				
	β-Caryophyllene			Control Plates	
	100%	10%	1%	Dimethyl Sulfoxide (0.5%)	Chloramphenicol (1 mg/mL)
MH	21 mm	11 mm	8 mm	6 mm	35 mm
MMD + glucose 1%	29 mm	17 mm	9 mm	7 mm	-
MMD	Absence of growth	Absence of growth	Absence of growth	Absence of growth	-

**Table 2 ijms-24-04845-t002:** Cross-streaking results. The different inhibition levels were indicated as follows: complete (4), strong (3), moderate (2), weak (1), and absent (0) inhibition.

Target		Inhibition Level	Target		Inhibition Level
	Strain			Strain	
CoNS	5419	0	*S. aureus*	ATCC 25923	0
	5318	0		2668	1
	5377	0		5788	0
	5403	0		3709	0
	5323	0		3710	0
	5321	2		4070	0
	5285	0		4168	0
	5284	0		4302	1
	5383	0		4691	1
	5396	0		4708	0
*P. aeruginosa*	ATCC 27853	1	*K. pneumoniae*	ATCC 700603	0
	5786	1		4409	2
	4189	1		4412	2
	5234	0		4417	2
	5245	0		4420	2
	7/1	1		4422	0
	7/2	0			
	7/3	0			
	5009	0			
	5236	0			

**Table 3 ijms-24-04845-t003:** General features of the genome of *Arthrobacter* sp. OVS8.

Attributes	Value
Genome size	4,175,013 bp
Contigs	1
G + C%	67.14%
Total genes	4273
CDS	2598
rRNA	15
tRNA	50
ncRNA	3
Accession number	CP116670

**Table 4 ijms-24-04845-t004:** Biosynthetic gene clusters identified using AntiSMASH. Nucleotide positions within the genome are reported.

BCG Type	From (nt)	To (nt)	Most Similar Known Cluster	Similarity
T3PKS	619,665	660,885	-	-
Betalactone	1,213,084	1,236,115	Microansamycin	7%
NAPAA	1,285,007	1,318,849	Stenothricin	31%
NRPS-independent-siderophore	2,670,131	2,682,100	Desferrioxamine	66%
NRPS	3,650,806	3,693,954	Skyllamycin D/skyllamicin E	5%

**Table 5 ijms-24-04845-t005:** Activity values (AV) of *Arthrobacter* sp. OVS8 on different antibiotics and chemical compounds. AV of the compounds used more efficiently (i.e., AV higher or equal to 7) and of those not used or inefficiently used (i.e., AV lower or equal to 4) are reported.

Well	Sensitivity Test	AV
A11	pH 6.0	9
B10	1% NaCl	9
B11	4% NaCl	9
C12	D-Serine	9
G10	Nalidixic Acid	9
G11	Lithium Chloride	9
G12	Potassium Tellurite	9
H10	Aztreonam	9
H12	Sodium Bromate	9
C10	1% Sodium Lactate	8
A12	pH 5.0	4
C11	Fusidic Acid	4
E12	Niaproof 4	4
F11	Tetrazolium Violet	4
F12	Tetrazolium Blue	4
B12	8% NaCl	2
D10	Troleandomycin	2
D11	Rifamycin SV	2
D12	Minocycline	2
E10	Lincomycin	2
E11	Guanidine hydrochloride	2
F10	Vancomycin	2

**Table 6 ijms-24-04845-t006:** Activity (AV) of *Arthrobacter* sp. OVS8 on different C sources. AV of the compound used more efficiently (i.e., AV higher or equal than 7), not used or inefficiently used (i.e., AV lower or equal than 3) and those modestly used (i.e., AV higher than 3 but lower than 7) are reported.

Efficiently Utilized C Sources		Inefficiently Utilized C Sources		Modestly Utilized C Sources	
Compound	AV	Compound	AV	Compound	AV
Beta-Hydroxy-D,D-Butyric Acid	9	Bromosuccinic Acid	3	Glycerol	6
D-Sorbitol	9	L-Alanine	3	Acetic Acid	6
D-Cellobiose	8	L-Histidine	3	D-Galactose	6
Sucrose	8	Citric Acid	3	L-Pyroglutamic Acid	6
D-Turanose	8	L-Fucose	2	Pyruvic Acid methyl ester	6
Stachyose	8	L-Arabitol	2	L-Malic Acid	6
D-Raffinose	8	D-Glucose-6-Phosphate	2	Gelatin	5
D-Melibiose	8	D-Serine	2	D-Fructose-6-Phosphate	5
Alpha-D-Glucose	8	3-Methyl-D-Glucoside	2	Quinic Acid	5
D-Mannose	8	L-Galactonic Acid-g-Lactone	2	L-Serine	5
Propionic Acid	8	D-Fucose	2	Pectin	5
L-Rhamnose	8	Sodium Butyrate	2	Dextrin	5
D-Trehalose	8	Mucic Acid	2	Acetoacetic Acid	4
D-Mannitol	8	D-Lactic Acid Methyl Ester	2	D-Saccharic Acid	4
Myo-Inositol	8	Tween 40	2	D-Glucuronic Acid	4
D-Maltose	8	Alpha-Ketoglutaric Acid	2	L-Glutamic Acid	4
D-Gluconic Acid	8	D-Malic Acid	2	Inosine	4
4-Hydroxyphenyl Acetic Acid	8	N-Acetyl-D-Galactosamine	1	Gentiobiose	4
L-Lactic Acid	8	Glucuronamide	1		
Alpha-Hydroxy-Butyric Acid	8	3-Methyl glucose	1		
L-Arginine	7	N-Acetyl-Neuraminic Acid	1		
Alpha-Keto-Butyric Acid	7	N-Acetyl-D-Mannosamine	1		
D-Fructose	7	D-Salicin	1		
		Formc Acid	1		
		N-Acetyl-D-Glucosamin	0		
		Alpha-D-Lactose	0		
		D-Galacturonic Acid	0		

**Table 7 ijms-24-04845-t007:** Pathogens’ antimicrobial resistance profile.

Pathogen	Strain	Antibiotic Resistance ^1^
CoNS	5419	FOX, DA, CIP, LEV, SXT, TIG
	5318	P, E, CN, FD
	5377	P, TE, E, TEC
	5403	P, E, TIG, TE
	5323	P, TE, TIG, E, CN
	5321	P, E, CN, AK, FD
	5285	E, CN, CIP, LEV, FD
	5284	P, TE, E, CN, FD
	5383	P, FOX, TE, E, CN
	5396	P, FOX, SXT, CN, FD
*P. aeruginosa*	ATCC 27853	FOX, K
	5786	CAZ, FEP
	4189	AK, TOB, CIP, LEV, CAZ, FEP, MEM, IPM, PRL, TZP
	5234	AK, CAZ, ATM, TZP, PRL, FEP, CN, IPM, MEM, LEV, CIP, TOB
	5245	CAZ, ATM, PRL, FEP, CN, LEV, CIP, IPM, MEM, TOB
	7/1	CAZ, FEP, MEM
	7/2	CAZ, FEP, MEM
	7/3	CAZ, ATM, FEP, MEM
	5009	ATM, CAZ, CIP, CN, FEP, IPM, LEV, MEM, PRL, TOB, TZP
	5236	AK, ATM, CAZ, CIP, FEP, IPM, LEV, MEM, TOB
*S. aureus*	ATCC 25923	P, NA
	2668	AMP, P, DA, TE, E
	5788	P, FOX
	3709	DA, TE, E
	3710	DA, TE, E
	4070	P, DA, TE, E, CIP, LEV, DAP
	4168	AMP, P, DA, SXT, TE
	4302	P, FOX, SXT, DAP
	4691	P, FOX, E, CN, CIP, LEV, DAP
	4708	P, FOX, CN, VA, DAP
*K. pneumoniae*	ATCC 700603	CAZ, AMP, ATM, PRL, TE
	4409	AK, AMX, FEP, CTX, CAZ, CIP, ETP, IPM, MEM, TZP, SXT, TIG
	4412	AMX, FEP, CTX, CAZ, CIP, ETP, IPM, MEM, TZP, SXT, TIG
	4417	AK, AMX, FEP, CTX, CAZ, CIP, ETP, IPM, MEM, TZP, SXT, TIG
	4420	AK, AMX, FEP, CTX, CAZ, CIP, IPM, MEM, TZP, SXT, CN
	4422	AK, AMX, FEP, CTX, CAZ, CIP, ETP, IPM, MEM, TZP, SXT

^1^ AK: Amikacin; AMX: Amoxicillin; AMP: Ampicillin; ATM: Aztreonam; CTX: Cefotaxime; CAZ: Ceftazidime; CIP: Ciprofloxacin; CN: Gentamicin; DA: Clindamycin; DAP: Daptomycin; ETP: Ertapenem; E: Erythromycin; FD: Fusidic acid; FEP: Cefepime; FOX: Cefoxitin; IPM: Imipenem; K: Kanamycin; LEV: Levofloxacin; MEM: Meropenem; NA: Nalidixic acid; P: Penicillin G; PRL: Piperacillin; SXT: Sulfamethoxazole/trimethoprim; TE: Tetracycline; TEC: Teicoplanin; TIG: Tigecycline; TOB: Tobramycin; TZP: Piperacillin/tazobactam; VA: Vancomycin.

## Data Availability

The data presented in this study are openly available in GenBank at https://www.ncbi.nlm.nih.gov/nuccore/2438204944, accessed on 30 January 2023, reference number CP116670.1.

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
