# Peer review of "Genomic, Molecular, and Phenotypic Characterization of Arthrobacter sp. OVS8, an Endophytic Bacterium Isolated from and Contributing to the Bioactive Compound Content of the Essential Oil of the Medicinal Plant Origanum vulgare L."

_ijms, 2023, doi:10.3390/ijms24054845_

Round 1

Reviewer 1 Report

The review of the manuscript ijms-2230444

The manuscript titled „Genomic, molecular, and phenotypic characterization of Ar- 2 throbacter sp. OVS8, an endophytic bacterium isolated from 3 the medicinal plant Origanum vulgare L., contributing to the bi- 4 oactive compounds content of the essential oil”,

In my opinion the gratest atribute of this worki s the fact that it presents the endophytic baxcterium as an antimicrobial secondary metabolite producers, especially VOC producers, since this subject is not studied well enough.

In general, the manuscript presents a wide array of microbiological and genetical mathods that have been properly used to characterie the Arthrobacter sp. OVS8 strain, in coordinance with the aim of the papaer.

This manuscript is worth publishing especially as an example of fine scientific abilities of characterisinig of the microbial strains. However a conclusions drawn should be supported with empirical data, when it is desired, like in the case of the influence of the strain on the plant growth. I will specify it below in detail.

All in all, I recommend this paper for publishing after minor revision. My remarks are listed below.

Line 157: „The obtained data obtained are shown in Table 2” – correct the repetitions in this sentence.

Line 196: GenBank annotation pipeline. If you have a complete genome of the strain Arthrobacter sp. OVS8, why can`t you provide this strain with a species name?. You give only the genus name. Why?. Please explain.

Line 454: „Arthrobacter sp. OVS8 was tested against 36 pathogenic strains”. In what way was the antibiotic resistance measured?. Except of the phrase in citation I dind not fin dany more information in this subject?. All the antibiotics are present in the Biolog Gen III Eco plates?. Please explain.

Line 459: „Tables may have a footer” please remove it. This is probably for the Journal`s template.

Lines 597-599: The aim of the work. In conclusions you write that the „ … this work aimed at characterizing the newly isolated endophytic strain Arthrobacter sp. OVS8, isolated from the stem of the medicinal plant O. vulgare to test its ability to influence plant growth and to synthesize antimicrobial compounds effective against bacterial human pathogens.”. But you did not provide the evidence for the actual influence of the OVS8 strain on the plant growth”. Correct me if I am wrong, but if not please rephrase this issue throughout the test (abstract, introduction, conclusion).

Kind regards,

The Reviewer

Reviewer 2 Report

See attached file
